# Homeodomain protein Otp affects developmental neuropeptide switching in oxytocin neurons associated with a long-term effect on social behavior

Einav Wircer[1], Janna Blechman[1], Nataliya Borodovsky[1], Michael Tsoory[2], Ana Rita Nunes[1,3], Rui F Oliveira[3,4], Gil Levkowitz[1]*

[1]Department of Molecular Cell Biology, Weizmann Institute of Science, Rehovot, Israel; [2]Department of Veterinary Resources, Weizmann Institute of Science, Rehovot, Israel; [3]Integrative Behavioural Biology Lab, Instituto Gulbenkian de Ciência, Oeiras, Portugal; [4]ISPA- Instituto Universitário, Lisboa, Portugal

**Abstract** Proper response to stress and social stimuli depends on orchestrated development of hypothalamic neuronal circuits. Here we address the effects of the developmental transcription factor orthopedia (Otp) on hypothalamic development and function. We show that developmental mutations in the zebrafish paralogous gene *otpa* but not *otpb* affect both stress response and social preference. These behavioral phenotypes were associated with developmental alterations in oxytocinergic (OXT) neurons. Thus, *otpa* and *otpb* differentially regulate neuropeptide switching in a newly identified subset of OXT neurons that co-express the corticotropin-releasing hormone (CRH). Single-cell analysis revealed that these neurons project mostly to the hindbrain and spinal cord. Ablation of this neuronal subset specifically reduced adult social preference without affecting stress behavior, thereby uncoupling the contribution of a specific OXT cluster to social behavior from the general *otpa*$^{-/-}$ deficits. Our findings reveal a new role for Otp in controlling developmental neuropeptide balance in a discrete OXT circuit whose disrupted development affects social behavior.

*For correspondence: gil. levkowitz@weizmann.ac.il

**Competing interests:** The authors declare that no competing interests exist.

## Introduction

The hypothalamus regulates homeostasis by receiving inputs from the internal and external environments and responding accordingly by the activation of neuro-endocrine and behavioral outputs (*Saper and Lowell, 2014*). Hypothalamus regulated processes include proper responses to anxiogenic and to social stimuli, which affect the animal's fitness. The development of the circuitry underlying hypothalamic functions is a highly complex process, which relies on orchestrated expression of transcription factors (*Puelles and Rubenstein, 2015*; *Domínguez et al., 2015*; *Machluf et al., 2011*). In humans, defects in hypothalamic development may lead to pathology (*Caqueret et al., 2005*). In particular, developmental disruptions of the oxytocin (OXT) system have been implicated in many pathological conditions, including autism and Prader-Willi syndrome, which are associated with impaired responses to stressful, social and metabolic stimuli (*Atasoy et al., 2012*; *Swaab et al., 1995*; *Lerer et al., 2008*; *Thompson et al., 2011*).

Despite the genetic associations between the OXT system and human diseases, the exact mechanism by which changes in the hypothalamic developmental plan affect behavior is not well understood. In this regards, relatively minor changes in gene expression during development may affect hypothalamic oxytocinergic (OXT-ergic) outputs. Such developmental variations in expression levels

**eLife digest** The life of most animals sees them encounter stressful situations and requires them to interact socially with other animals. A region of the brain called the hypothalamus controls the behavioral response to social and stressful situations. In humans, defects that affect how neurons in the hypothalamus develop have been linked to autism and other behavioral disorders.

In many animal species, a protein called orthopedia is essential for the neurons in the hypothalamus to develop and work properly and mammals that lack the gene to make this protein do not make it past birth. Zebrafish have two closely related genes that code for orthopedia, referred to as *otpa* and *otpb*. That means that fish bearing a mutation in either one of these genes can be used to investigate its physiological effects in adult animals.

Wircer et al. have now used genetic tools to investigate how orthopedia affects the ability of zebrafish to respond to social and stressful situations. It was found that zebrafish with a mutant form of *otpa* – but not those animals with a mutant form of *otpb* – display anxiety-like behavior when faced with a stressful situation. These fish also show abnormal social behavior, displaying measurably decreased tendencies to swim in a 'social zone' – an area next to a visible tank compartment that contains a school of zebrafish.

Further investigation linked the social preferences of the fish to a particular circuit of neurons that produce the neurotransmitter oxytocin, which is known to affect social affiliation in many species. Investigation of other neurotransmitters revealed that these particular neurons also produce corticotropin-releasing hormone, which is known to regulate the response to anxiety and stress. Wircer et al. found that orthopedia regulates how much of each neurotransmitter is coproduced by the same neurons. This ability to change the balance of neurotransmitter production may allow the fish to switch between the "social" and "stress" states, enabling them to rapidly adapt to environmental changes and change their behavior.

Exactly how orthopedia regulates the balance of neuropeptide production – and how this influences behavior – remains a question to be answered by further studies. More work is also needed to determine how these results relate to what occurs in the brains of mammals.

of OXT and/or its cognate receptor are not necessarily lethal; however, they may disrupt both physiological and psychological responses such as stress and social behaviors (*King et al., 2016*; *Bosch et al., 2005*). Similarly, genetic variations in the V1a receptor for arginine-vasopressin (AVP), which often functionally synergizes with or antagonizes OXT signaling, are associated with changes in personality features in chimpanzees as well as pair bonding behavior in humans (*Walum et al., 2008*; *Hopkins et al., 2012*).

The possible link between the activity of critical transcriptional regulators of hypothalamic development and long-term effects on the animal's ability to respond to homeostatic challenges has been discussed [reviewed in *Biran et al., 2015*)]. Works of several labs including ours have focused on the homeodomain transcription factor orthopedia (Otp), which is essential for proper hypothalamic differentiation. Otp is important for the development of hypothalamic neurons, including OXT neurons in mouse (*Acampora et al., 1999*; *Wang and Lufkin, 2000*) and fish (*Eaton et al., 2008*; *Blechman et al., 2007*; *Ryu et al., 2007*). In addition to its effect on hypothalamic development, zebrafish Otp mutants display impaired behavioral response to homeostatic challenges such as adaptation to novel environment as well as dark-induced photokinesis (*Amir-Zilberstein et al., 2012*; *Fernandes et al., 2012*). Given the role of Otp in the development of OXT neurons and the established role of OXT in the regulation of social behavior across vertebrates, it is expected that Otp may not only be involved in the stress response but also to have a developmental effect on social behavior.

In the present work we addressed the long-term effect of Otp on hypothalamic functions by examining the consequences of developmental mutations of the two zebrafish paralogs, *otpa* and *otpb*, on the behavioral responses of these mutants to anxiogenic and social challenges. We show that *otpa*, but not *otpb*, mutants display anxiety-like and social-related defects. Subsequently we demonstrated that the two *otp* genes differentially regulate the expression of OXT (a.k.a. isotocin in

fish) and corticotropin-releasing hormone (CRH) in a newly identified OXT neuronal cluster. Finally, we show that these OXT cells, which project mainly to the hindbrain and spinal cord, are associated with the modulation of social behavior, but not with the response to stressful stimuli.

## Results

### otpa but not otpb mutants display anxiety and social behavior deficits

In mouse, *Otp* knockout results in early lethality of the null pups (*Acampora et al., 1999*; *Wang and Lufkin, 2000*), which hinders the investigation of long-term effects on adult physiological function. Zebrafish express two paralogous genes, namely *otpa* and *otpb*, whose expression patterns largely overlap. Thus, adult fish with a single mutation in either gene are viable and fertile (*Ryu et al., 2007*; *Fernandes et al., 2013*; *Amir-Zilberstein et al., 2012*). To examine the consequences of developmental mutations of both *otpa* and *otpb* on adult physiology, we tested the behavioral responses of these mutant fish to stressful and social challenges. These challenges are known to trigger evolutionarily conserved behaviors that depend on normal hypothalamic development (*Biran et al., 2015*; *Szarek et al., 2010*; *Carter, 2003*).

o*tpa* and *otpb* mutants were tested in two behavioral paradigms that aim to measure anxiety and social preference. The open field paradigm is a standard anxiety-like behavioral test in rodents that has been adapted to zebrafish (*Champagne et al., 2010*; *Nunes et al., 2016*). In this test adult zebrafish are transferred to a novel circular arena where time spent near the walls (thigmotaxis) is taken as an anxiety-like behavior, since anxiolytic drugs can shift this preference (*Cachat et al., 2010*; *Kalueff et al., 2013*; *Schnörr et al., 2012*). We examined the location and velocity of the fish during the whole trial (10 min). Upon introduction into the novel environment, wild type fish displayed a preference to remain in the center of the arena swimming at a relatively low speed (*Figure 1A,B*). Over time, these fish gradually swam towards the walls and increased their swimming velocity until they reached a steady-state average distance of 1–2 cm from the walls, with an average speed of ~10 cm/sec. $otpb^{-/-}$ mutants displayed similar swimming patterns to those of wild type fish (*Figure 1A,B*). In contrast to both, the $otpa^{-/-}$ mutant displayed significantly unusual swimming patterns (*Figure 1A,B*). Throughout the test, $otpa^{-/-}$ fish tended to freeze more, spent most of the time near the center of the arena, and their average speed did not increase (*Figure 1A,B* and *Figure 1—figure supplement 1*). Notably, no difference was observed in the locomotor activity of $otpa^{-/-}$ mutants versus wild types or $otpb^{-/-}$ when tested in their home tank environment. This indicates that the open field phenotype is not due to motor deficits and suggests a context-dependent behavior of the mutant (*Figure 1C,D*). These results indicate that $otpa^{-/-}$ but not $otpb^{-/-}$ mutants display aberrant response to an open field consistent with the previously established role of Otpa in stress adaptation (*Amir-Zilberstein et al., 2012*).

Next, we examined the response of $otpa^{-/-}$ and $otpb^{-/-}$ mutants to social stimuli by employing a visually-mediated social preference (VMSP) paradigm. This test has been widely used to measure zebrafish sociality, by quantifying the zebrafish preference to associate with a shoal, i.e. to swim next to a group of conspecifics (*Fernandes et al., 2015*; *Engeszer et al., 2007*; *Nunes et al., 2016*). For that, we have devised a behavioral arena of three compartments separated by transparent barriers, in which the focal fish can choose to either roam the large part of the arena or swim near the two other compartments. One compartment contained a group of four fish (i.e. shoal), and the region of interest next to it was termed the 'social zone'. The second compartment was empty and the adjacent region of interest was termed the 'non-social' zone (*Figure 1E*). Upon introduction to the main compartment, both wild type and $otpb^{-/-}$ mutant fish rapidly entered the 'social zone' and spent a substantial portion of their time next to the shoal compartment rather than in the 'non-social zone' (*Figure 1F,G*). In comparison, $otpa^{-/-}$ mutants spent significantly less time in the 'social zone' (*Figure 1F*). Given the result of the open field *otpa* mutants might be less likely to enter the social zone due to increased anxiety. To address this, we analyzed the number of entries into the social zone as well as the habituation to the preference arena as measured by the distance moved and speed over time. We found no significant difference in all three parameters between wild type and *otpa* mutant animals (*Figure 1—figure supplement 1*). We conclude that *otpa* mutants display normal habituation to the VMSP arena as well as exploration of the social zone.

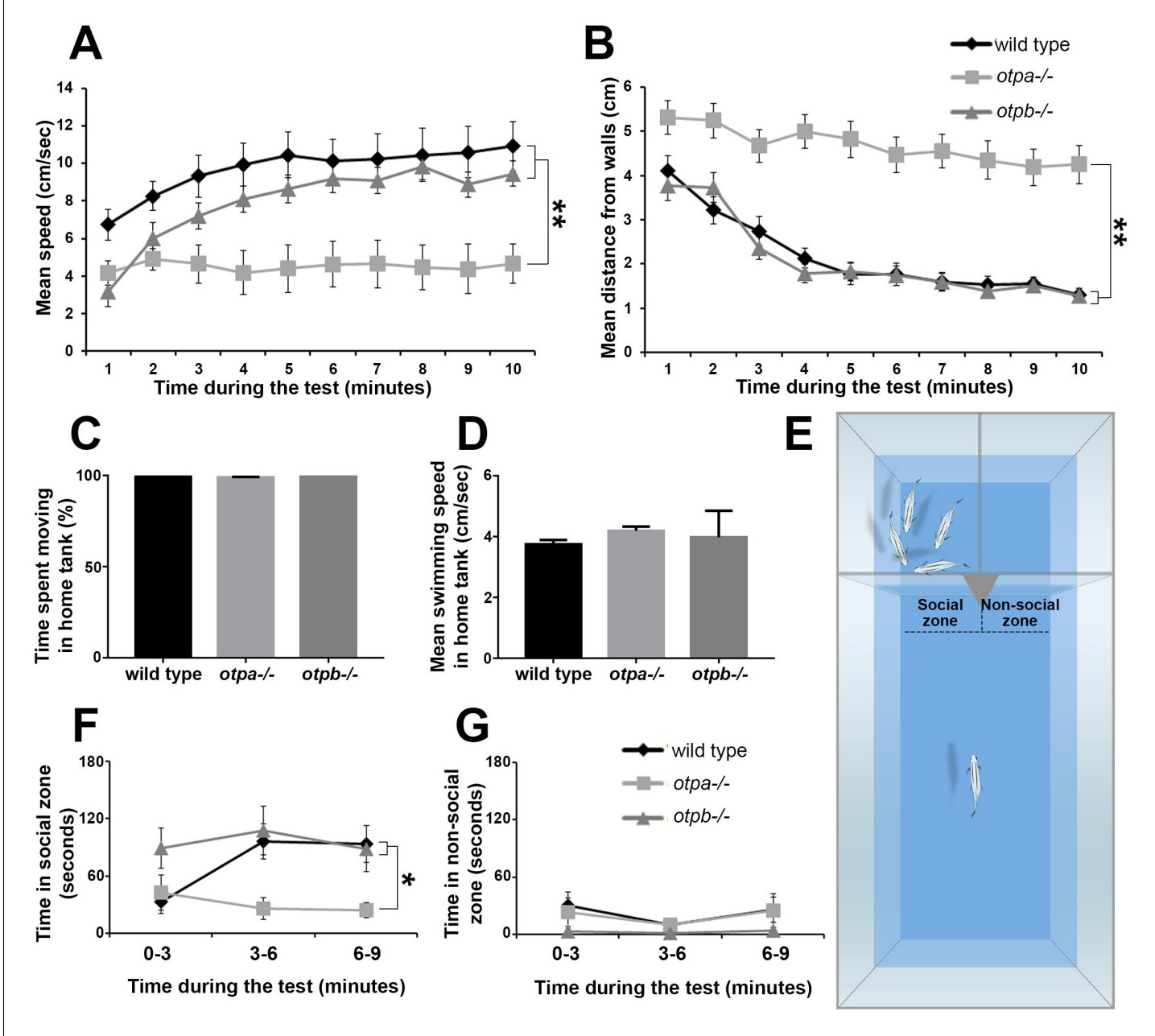

**Figure 1.** *otpa* but not *otpb* mutants display altered stress and social behavioral responses. (A,B) Fish were placed in a novel circular tank and their behavior was recorded for 10 min. The mean and standard error (SEM) of the swimming speed (A) and the distance from the arena walls (B) in every minute of the test duration are plotted for wild types (*n* = 12), *otpa*$^{-/-}$ mutants (*n* = 23) and *otpb*$^{-/-}$ mutants (*n* = 13). Both wild type and *otpb*$^{-/-}$ exhibited significant (\*\*p<0.01) habituation, as manifested by increased speed (A) and decreased distance from the wall (B), whereas *otpa*$^{-/-}$ did not habituate to the arena; they swam significantly (\*\*p<0.01) slower (A) and farther from the wall (B) than both wild type and *otpb*$^{-/-}$ throughout the session. (C,D) Wild types (*n* = 12), *otpa*$^{-/-}$ (*n* = 20) and *otpb*$^{-/-}$ (*n* = 4) were recorded swimming for one minute in their home tanks and their locomotor parameters were analyzed. No differences were observed between the genotypes in either percent of time spent moving [X$^2$$_{(2)}$=3.605; p=0.165; C] or swimming speed [F$_{(2)}$=1.293; p=0.288; D]. (E–G) Visually mediated social preference behavioral test of wild type, *otpa*$^{-/-}$ and *otpb*$^{-/-}$ mutants (*n* = 10 each). Fish were put in an isolated arena from which two compartments were visible, one containing a four-fish shoal and the other was empty (E). The time spent next to the shoal ('*social zone*'; F) and next to the empty compartment ('*non-social zone*'; G) were measured in three-minute time bins. Both wild type and *otpb*$^{-/-}$ exhibited significantly (\*p<0.05) different social preference (F) specifically in the second and third time bins. Similar analyses of '*time spent in the non-social zone*' indicated no differences between the genotypes (G).

The following source data and figure supplements are available for figure 1:

*Figure 1 continued on next page*

*Figure 1 continued*

**Source data 1.** Swimming parameters for the behavioral tests.
**Figure supplement 1.** *otpa*$^{-/-}$ fish freeze more in the open field arena but display normal exploration of the social zone.
**Figure supplement 1—source data 1.** Swimming parameters for the behavioral tests.

Taken together, the behavioral results suggest that in zebrafish, *otpa* is necessary for proper responses to both anxiogenic and social stimuli in a manner that is not redundant with its paralogous gene *otpb*.

## Otpa represses OXT expression in a newly identified cluster of parvocellular OXT neurons

OXT is a neuropeptide that is associated with pro-social and anxiolytic activities (*Burkett et al., 2016*; *Dölen et al., 2013*). Therefore, we inquired whether the behavioral deficits of Otp mutant might be due to developmental impairments in OXT neurons. Notably, a previous study reported that single mutations in either *otpa* or *otpb* had no significant effect on the number of OXT cells (*Fernandes et al., 2013*). Indeed, the results of in situ hybridization of *oxt* mRNA were in agreement with that report, showing that the number of cells in the main cluster of OXT neurons, which resides in the zebrafish neurosecretory preoptic area (NPO), is not affected in *otpa*$^{-/-}$ and *otpb*$^{-/-}$ mutants (*Figures 2* and 4E). However, careful examination of *otpa*$^{-/-}$ mutants revealed that starting from day three post fertilization (three dpf), a small group of cells in the posterior tuberculum (PT) region expressed *oxt* mRNA (*Figure 2D* and *Figure 2—figure supplement 1*).

Because an OXT neuronal cluster residing in the PT has never been reported, we sought to determine whether the observed phenotype reflected a mutant anomaly by searching for OXT-expressing neurons in the PT of wild type animals. We had previously generated a transgenic oxt:EGFP reporter [*Tg(oxt:egfp)*], which faithfully represents the endogenous expression of *oxt* mRNA and protein (*Blechman et al., 2011*; *Gutnick et al., 2011*). We noticed the existence of a small cluster of EGFP-expressing cells in the PT of the *Tg(oxt:egfp)* reporter (*Figure 2B*), whose anatomical location and arrangement resembled the presumably ectopic OXT cells observed in the *otpa*$^{-/-}$ mutant. We therefore crossed *Tg(oxt:egfp)* with *otpa*$^{-/-}$ zebrafish and examined whether the ectopic OXT-positive neurons co-localized with transgenic EGFP-positive neurons. Indeed, the EGFP-positive cells in the PT expressed high levels of *oxt* mRNA in the *otpa*$^{-/-}$ but not in wild type animals (*Figure 2C,F*).

Based on these results, we hypothesized that wild type fish might contain genuine OXT neurons in the PT, which express nearly undetectable levels of OXT due to a repressor activity of Otpa. These PT OXT neurons are visible in the transgenic reporter probably due to the absence of a genomic repressor element in the reporter. To test this hypothesis, we used a highly sensitive fluorescent in situ hybridization (FISH) method (*Orjalo et al., 2011*). The approach is based on a set of FISH probes comprising multiple oligonucleotides with different sequences and fluorescent labels, which collectively bind along the same target transcript to produce a signal of single-molecules. Using this method, the expression of *oxt* mRNA in PT neurons could be detected in wild type animals (*Figure 2G–I*). Furthermore, wild type embryos that were subjected to in situ hybridization with a DIG-labeled *oxt* probe followed by long incubation with the NBT/BCIP colorimetric substrate also revealed low level of *oxt* mRNA in PT neurons (*Figure 2—figure supplement 2A*). Moreover, we found that the PT OXT neurons express the oxytocin receptor (OXTR) (*Figure 2—figure supplement 2B*). Because OXT neurons are distinguished by the presence of somatodendritic autoreceptors (*Freund-Mercier and Stoeckel, 1995*), this finding evidences that the seemingly ectopic cells we uncovered by combining the *otpa* mutant with a transgenic OXT reporter are *bona fide* OXT neurons.

We hypothesized that the newly identified PT OXT neurons could be very similar to parvocellular OXT neurons in mammals (*Knobloch and Grinevich, 2014*). To address this issue, we measured OXT neurons soma size in both larval (5 day-old) and adult zebrafish (*Figure 3*). We observed that

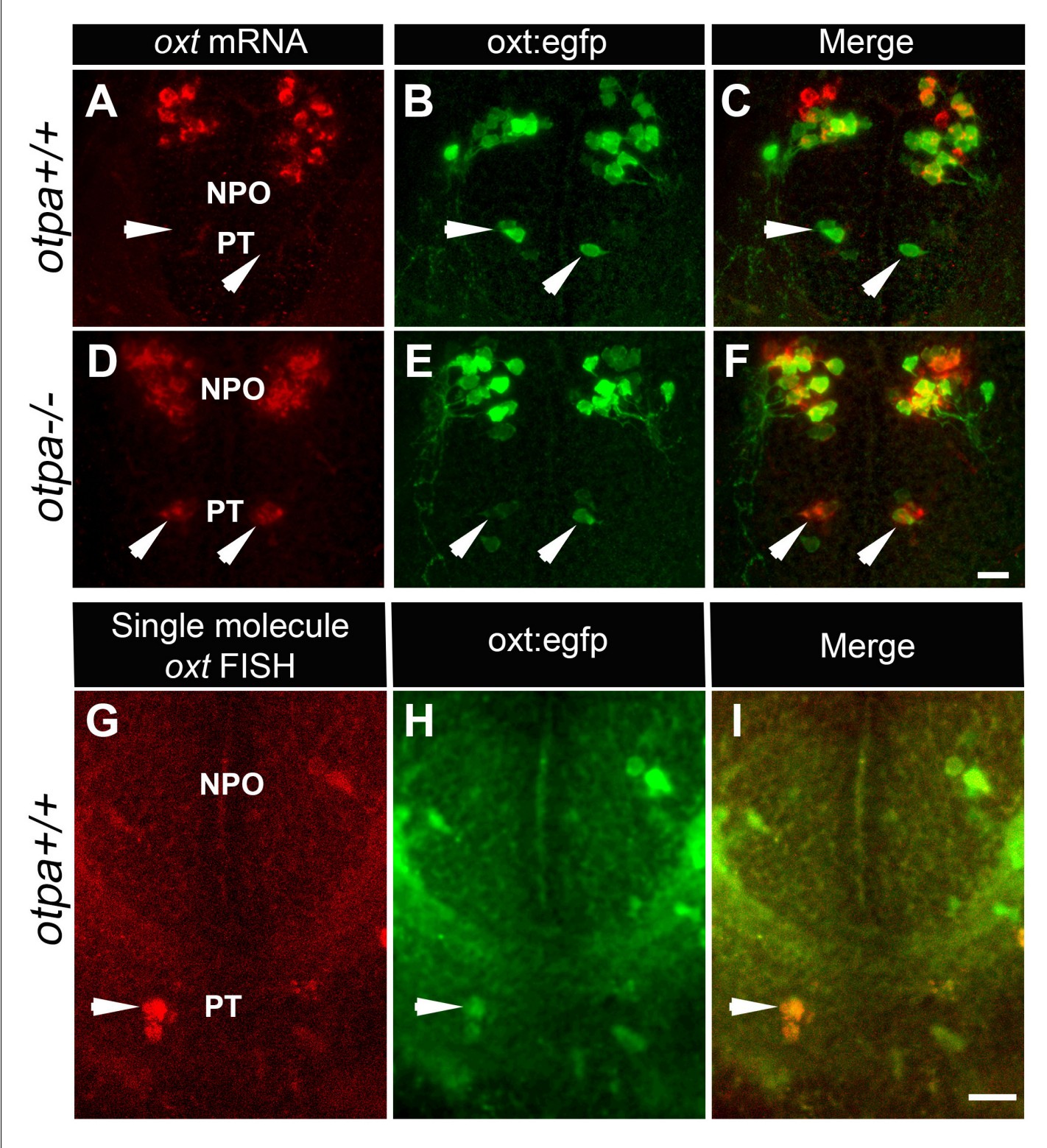

**Figure 2.** *otpa* mutant reveals a new cluster of hypothalamic OXT neurons. (**A–F**) In situ hybridization of *oxt* mRNA in 5 day-old wild type (*otpa*$^{+/+}$) and *otpa*$^{-/-}$ mutant on the background of a transgenic OXT reporter *Tg(oxt:egfp)* followed by confocal imaging (dorsal view, anterior to the top). *otpa*$^{+/+}$; *Tg(oxt:egfp)* fish do not express detectable levels of *oxt* mRNA (**A**), but express EGFP in the posterior tuberculum (PT; **B**). *otpa*$^{-/-}$; *Tg(oxt:egfp)* mutants express *oxt* mRNA (**D**) in *oxt:egfp*-positive cells of the PT (**E**). PT OXT neurons are indicated by arrowheads. Scale bar, 20 μm. (**G–I**) A representative image (single confocal plane) of a transgenic [*otpa*$^{+/+}$;*Tg(oxt:egfp)*] larvae, which was subjected to fluorescent in situ hybridization (FISH) with a single-

*Figure 2 continued on next page*

*Figure 2 continued*

molecule *oxt* mRNA probe (Stellaris) showing co-localization (arrowhead) of *oxt* in EGFP-labelled PT cells. NPO, neurosecretory preoptic area. Scale bar, 20 μm.

The following figure supplements are available for figure 2:

**Figure supplement 1.** Ectopic *oxt* mRNA expression in *otpa*$^{-/-}$ mutants.

**Figure supplement 2.** Expression of *oxt* and its receptor by PT OXT neurons.

**Figure supplement 3.** Posterior tuberculum (PT) OXT neurons express *otpa* and *otpb*.

the larval PT OXT neurons were intermingled with the pear-shaped tyrosine hydroxylase (TH)-positive dopaminergic neurons (a.k.a group no. 2) that have been well characterized in both larval and adult diencephalon (*Rink and Wullimann, 2002*). We therefore used TH immunoreactivity as an anatomical landmark to localize the PT OXT neurons in the adult periventricular PT (*Figure 3*). Our results indicate no major difference in cell size between the NPO and PT OXT groups in the larvae. However, in the adult brain, the PT OXT neurons were similar in their size to the anterior parvocellular preoptic nucleus and were significantly smaller than the known zebrafish magnocellular and gigantocellular neurons (*Figure 3*).

Taken together, these results revealed a new OXT-ergic neuronal cluster of zebrafish parvocellular neurons, which resides in the diencephalic PT domain and is spatially distinct from the NPO parvocellular neurons. These neurons express low level of OXT that is negatively regulated by Otpa. The expression of OXT in the PT is unleashed in the *otpa*$^{-/-}$ mutant, leading to higher levels of *oxt* mRNA.

## Otp paralogs exert differential effects on OXT neuronal clusters

The NPO and PT co-express *otpa* and *otpb* (*Herget et al., 2014*) and both genes are expressed within OXT neurons residing in the PT and NPO (*Figure 2—figure supplement 3*). Therefore, we examined whether genetic interaction between *otp* paralogous genes might affect the two OXT neuronal clusters. The analysis revealed intricate genetic interactions between the two zebrafish Otp paralogs. Thus, in agreement with previous reports of developmental knockout of the single *otp* mouse ortholog (*Acampora et al., 1999*; *Wang and Lufkin, 2000*), no *oxt* mRNA was observed in either the NPO or PT areas of *otpa*$^{-/-}$;*otpb*$^{-/-}$ double mutant (*Figure 4D,E and F*). We also evidenced a slight, but significant reduction of OXT cell number in *otpa*$^{-/-}$;*otpb*$^{+/-}$, suggesting a gene dose effect (*Figure 4E*). Mutations in *otpa* or *otpb* alone did not result in a change in the number of OXT-ergic neurons in the NPO, presumably due to functional redundancy (*Figure 4B,C and E*). In contrast to the phenotype we observed in the *otpa*$^{-/-}$ mutant, Otpb loss-of-function did not affect the OXT neurons located in the PT, indicating differential effects of Otpa and Otpb in this newly identified OXT cluster (*Figure 4C,F*).

In view of the above results, we conclude that both Otpa and Otpb positively and redundantly regulate OXT expression in the NPO; yet, they have opposing effects on OXT cells in the PT. Whereas Otpb induces OXT expression in this region, Otpa negatively regulates the expression of this neuropeptide (see model in *Figure 5I*).

## Otp paralogs differentially regulate neuropeptide switching in OXT neurons

We next asked whether OXT neurons in the PT cluster have a unique molecular composition that differentiates them from cells in the anterior NPO. However, analysis of dozens of hypothalamic markers, including transcription factors and neuropeptides, failed to detect differential gene expression between the NPO and PT OXT neurons. During the analysis, we noticed that as reported in mammals (*Sawchenko et al., 1984*) and zebrafish (*Herget and Ryu, 2015*), a subset of zebrafish OXT neurons co-express both OXT and the stress neurohormone CRH (*Figure 5*). Therefore, we analyzed the expression of *crh* mRNA in OXT neurons in *otpa* and *otpb* mutants. In wild type embryos,

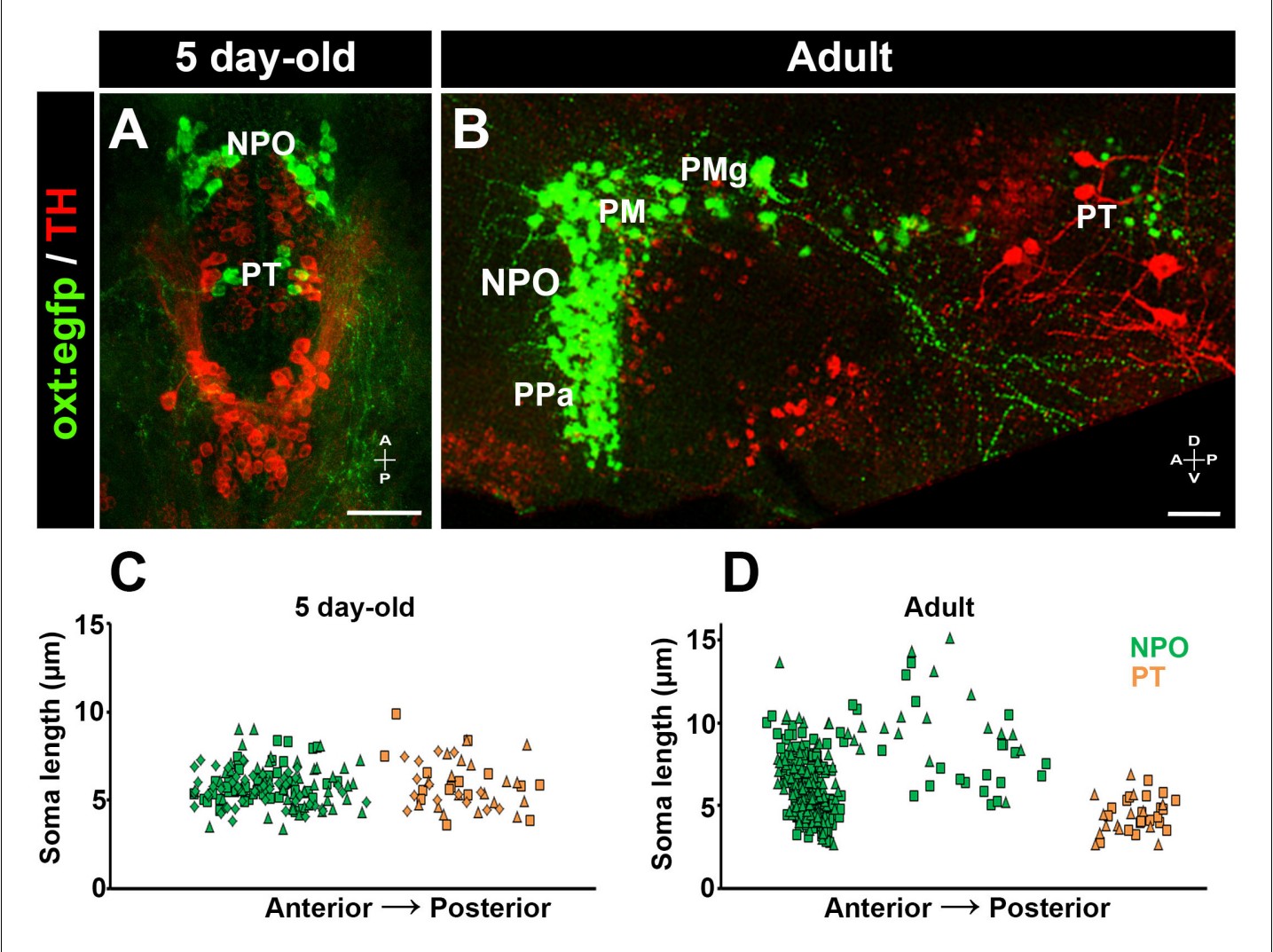

**Figure 3.** Posterior tuberculum (PT) OXT neurons represent a new cluster of zebrafish parvocellular neurons. Immunostaining and confocal imaging of either 5 day-old larva (A) or adult (B; 150 µm sagittal section) transgenic OXT reporter *Tg(oxt:egfp)* with a tyrosine hydroxylase (TH) antibody which serves as an anatomical landmark. The soma size of OXT neurons was measured using FIJI image-processing package in three larvae (C) and two adults (D) and plotted as a function of their relative anterior-posterior position. Each individual animal is marked by a different shape. NPO and PT neurons are labeled in green and orange, respectively. NPO, neurosecretory preoptic area; PM, magnocellular preoptic nucleus; PMg, gigantocellular part of magnocellular preoptic nucleus; PPa, parvocellular preoptic nucleus - anterior part; PT, posterior tuberculum; TH, tyrosine hydroxylase.

The following source data is available for figure 3:

**Source data 1.** Size and location of OXT cell bodies.

approximately 7–10% of OXT neurons in both the NPO and PT clusters co-expressed *crh* (*Figure 5G,H*). *otpa*[−/−] mutants, however, displayed an increase in *crh*-positive OXT neurons in the NPO and a decrease in the PT OXT neurons (*Figure 5B,E,G,H*). Moreover, similar analysis of *crh*-positive OXT neurons in the *otpb*[−/−] mutant indicated a positive trend for Otpb regulation of *crh* expression in the PT clusters of OXT neurons (*Figure 5F,H*). Notably, previous studies have shown that the total number of CRH+ neurons is not affected in either *otpa* or *otpb* mutants reinforcing the notion that the Otp paralogs regulate the balance between *crh* and *oxt* (*Fernandes et al., 2013*; *Amir-Zilberstein et al., 2012*).

In summary of the results shown in *Figures 4* and *5*, we propose that Otp paralogs induce differential and spatially dependent neuropeptide switching phenotypes in OXT neurons (*Figure 5I*).

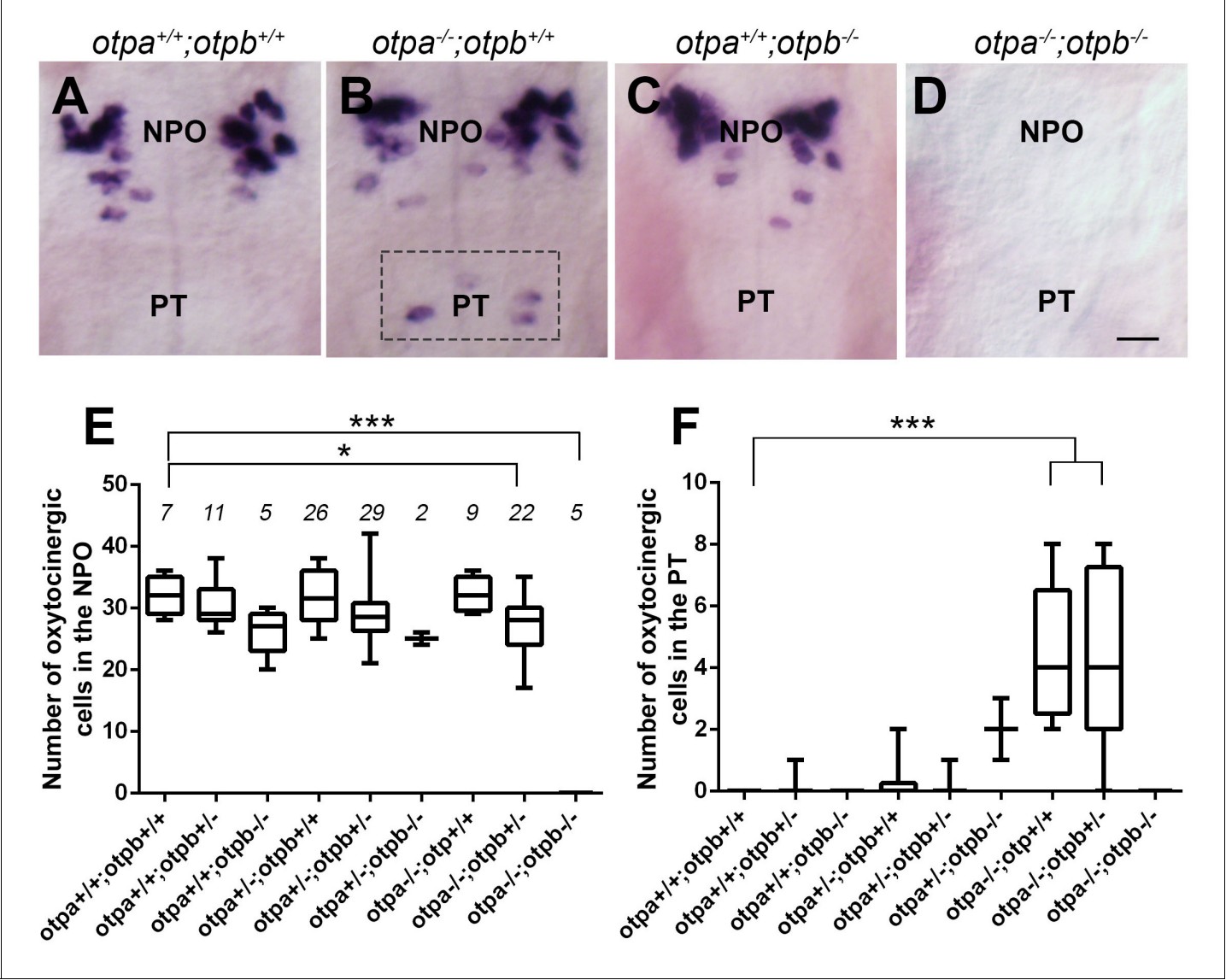

**Figure 4.** Otp paralogs have differential effect on hypothalamic *oxt* expression. (A–D) Representative high-resolution micrographs of 5 day-old embryos that were subjected to whole mount in situ hybridization with *oxt* mRNA probe (dorsal view, anterior to the top). *oxt*-positive neurons in the posterior tuberculum (PT) of *otpa*$^{-/-}$ mutants are marked by a dotted rectangle (B). Scale bar, 20 μm. (E,F) Box and whisker plots showing OXT cell number counts in the neurosecretory preoptic area (NPO; E) and in the PT (F) in various genotypes. The numbers of embryos used for the quantification are indicated above each box. In the NPO, *otpa*$^{+/+}$;*otpb*$^{+/+}$ fish differed significantly only from *otpa*$^{-/-}$;*otpb*$^{+/-}$ (*p=0.013) and *otpa*$^{-/-}$;*otpb*$^{-/-}$ (***p=0.000). In the PT *otpa*$^{+/+}$;*otpb*$^{+/+}$ differed significantly (***p=0.000) only from *otpa*$^{-/-}$;*otpb*$^{+/-}$ and *otpa*$^{-/-}$;*otpb*$^{+/+}$.

The following source data is available for figure 4:

**Source data 1.** Number of OXT neurons in the NPO and PT.

Thus, Otpa has opposite effects on the expression of *oxt* and *crh* in a subset of OXT neurons residing in the PT and NPO. Otpb, on the other hand, positively regulates these neuropeptides in the PT and has a redundant effect on *oxt* expression in the NPO. Whether or not Otp paralogs have a broader effect on gene expression patterns in discrete subsets of OXT neurons is yet to be determined.

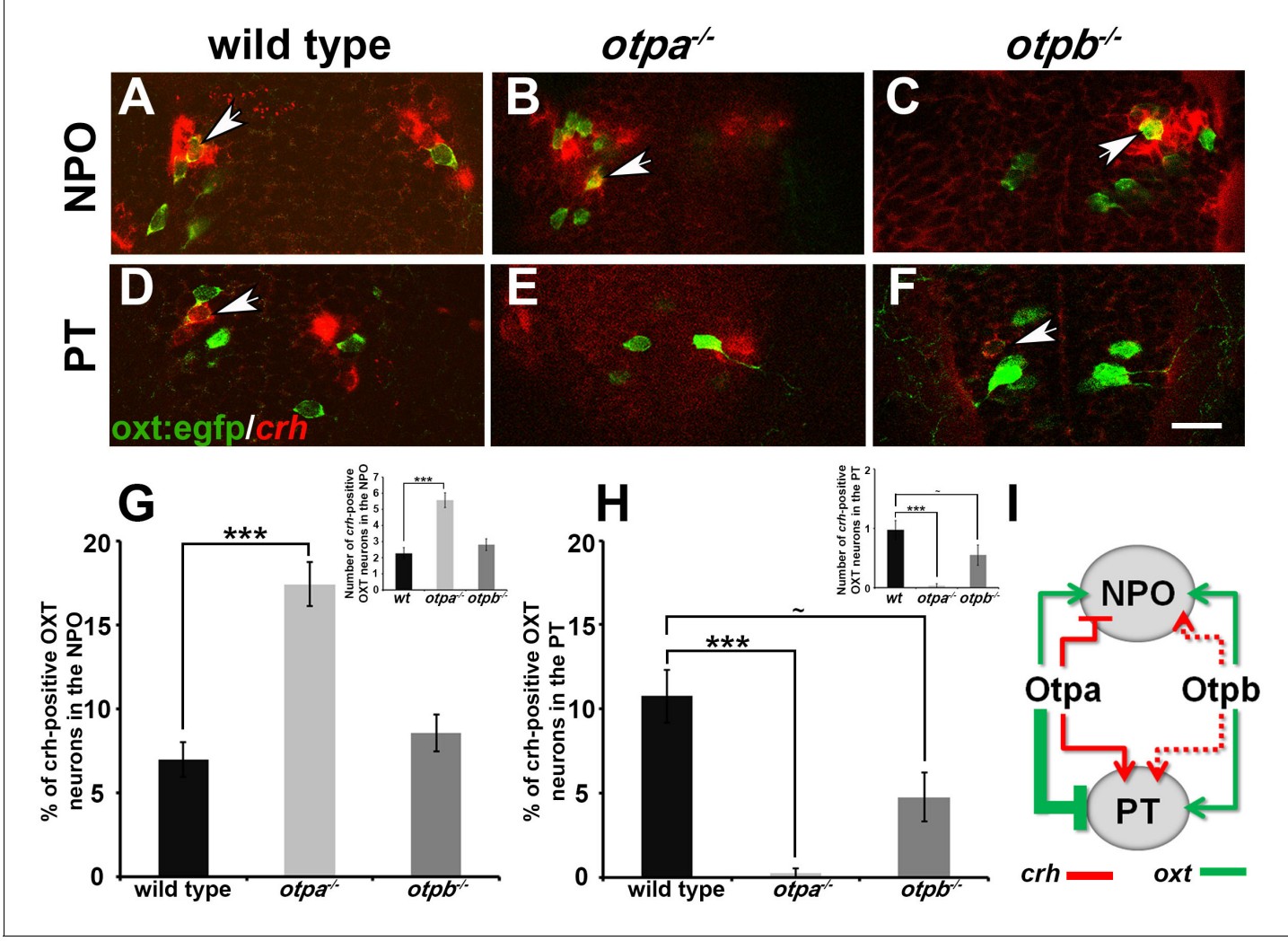

**Figure 5.** Otp paralogs regulate neuropeptide switching in OXT neurons. (A–F) In situ hybridization of *crh* mRNA in wild type, *otpa*$^{-/-}$ and *otpb*$^{-/-}$ on the background of a transgenic OXT reporter [*Tg(oxt:egfp)*]. The image panels show representative single confocal planes (dorsal view, anterior to the top) of OXT neurons in either the neurosecretory preoptic area (NPO; A–C) or the posterior tuberculum (PT; D–F). OXT neurons expressing *crh* mRNA are indicated by arrowheads. Scale bar, 20 μm. (G,H) Bar histogram showing the percentage (%) and cell count (upper right insets) of OXT cells co-expressing *crh* in the NPO (G) and PT (H). wild type (*n* = 40), *otpa*$^{-/-}$ (*n* = 30) and *otpb*$^{-/-}$ (*n* = 20). Kruskal-Wallis indicated a significant main effects for 'genotype' on the percentage of *crh*-positive OXT neurons in both the NPO [X$^2_{(2)}$=29.284; p=0.000] and PT [X$^2_{(2)}$=27.174; p=0.000]; Dunn's corrected pair-wise comparisons indicated that in both the NPO and PT, wild type differed significantly from *otpa*$^{-/-}$ (p=0.000). Notably, *otpb*$^{-/-}$ mutants exhibit a trend for decreased (p=0.077) *crh*-positive OXT neurons when compared to WT. (I) A model summarizing the suggested effects of otp paralogs on neuropeptide switching in OXT neuronal clusters in NPO and PT based on the results presented in *Figures 4* and *5* (see text). Arrows and T-bars indicated a positive and negative effect, respectively. Dotted arrows indicate a trend for a positive effect of *otpb* on *crh* expression.

The following source data is available for figure 5:

**Source data 1.** Percentage of CRH-positive OXT neurons in the NPO and PT.

## PT OXT neurons project mainly to the spinal cord

Proper connectivity is vital for the function of a neuronal system. Studies in mammals have shown that OXT neurons connect with different areas in the brain and spinal cord (*Dölen et al., 2013*; *Knobloch et al., 2012*; *Onaka et al., 2012*). In fish, the OXT-ergic system is far less characterized, and the correlation between neuronal nuclei in mammals and zebrafish is not fully understood (*Löhr and Hammerschmidt, 2011*).

To better understand the identity of the various OXT-ergic cells and, specifically, of the Otp-affected OXT neurons in the PT, we established a genetic labeling system that allowed us to trace the projections made by each OXT-ergic neuron at a single-cell resolution. To this end, we generated a transgenic construct encoding for the trascription activator GAL4 under the regulation of the *oxt* promoter. In addition, we have constructed a plasmid encoding for a membrane-bound *tRFP-caax* under a *UAS* promoter, which is activated by the GAL4 protein. The *oxt:Gal4* and *UAS:tRFP-caax* plasmids were injected into one-cell stage *Tg(oxt:egfp)* embryos, leading to tRFP labeling of the membranes in single OXT neurons (*Figure 6—figure supplement 1*). At 5 dpf, we stained the fish with antibodies to EGFP and tRFP and screened for single labeled neurons, which were detected in approximately 3–5% of the injected embryos. Using this method, we labeled 101 single OXT neurons, 56 in the NPO (*Figure 6*) and 36 in the PT (*Figure 7*). We also found 9 non-OXT-ergic cells, probably representing ~9% ectopic labeling (data not shown).

The analysis revealed six types of morphologically distinct neurons in the NPO (*Figure 6* and *Figure 7—figure supplement 1*): (1) hypothalamo-neurohypophyseal projections to the posterior pituitary (4/56); (2) neurons with local projections within the hypothalamus and NPO (3/56); (3) neuron with local commissures at the hypothalamus level (1/56); (4) neurons with local projections to the telencephalon (2/56); (5) neurons that project to the hindbrain (14/56); and (6) neurons with dual projections to the spinal cord and hypothalamus (32/56).

Importantly for this study, we have identified four types of projecting OXT neurons in the PT (*Figure 7* and *Figure 7—figure supplement 1*): (1) local commissural neurons (4/36); (2) neurons with hindbrain projections (7/36); (3) neurons with spinal cord projections without commissures (16/36); and (4) neurons with projections to the spinal cord and contralateral commissures (9/36). Notably, only NPO but not the PT OXT neurons displayed hypothalamo-neurohypophyseal projections that were previously shown to interface with the hypophyseal vasculature (*Gutnick et al., 2011*). Hence, the vast majority (~90%) of the PT OXT neurons shown to be affected by Otpa had descending projections to the hindbrain and spinal cord and did not form hypothalamo-neurohypophyseal neuroendocrine projections (*Figure 7G*).

## Early ablation of PT OXT neurons affects adult social but not stress related behavior

We demonstrated that $otpa^{-/-}$ mutants exhibit deficits in social- and anxiety-related behaviors, which are correlated with expression of OXT as well as neuropeptide switching in a new cluster of OXT neurons located in the PT. However, $otpa^{-/-}$ mutants have defects in the development of several types of neurons, such as dopaminergic cells (*Blechman et al., 2007*; *Fernandes et al., 2013*; *Ryu et al., 2007*) and deep brain photoreceptor cells (*Fernandes et al., 2012*). To explore how specific neuro-developmental changes in hypothalamic OXT neurons might affect behavioral functions, we focused on the PT OXT neuronal cluster for several reasons. First, the predominant $otpa^{-/-}$ mutant phenotype, i.e. ectopic OXT expression, was observed in these neurons. Second, unlike the dual central and peripheral (i.e. neurohypophyseal) connections made by the NPO OXT neurons, PT OXT neurons display a prevalent hindbrain and spinal cord projection pattern. Lastly, the small number of cells in the PT cluster renders them more amenable to perturbation.

To study the specific association of the PT OXT neuronal cluster with adult behavior, we used a two-photon laser microscope to photo-ablate these neurons at the larval stage, during which we had observed the Otpa-dependent neuropeptide phenotype. We next monitored the related behavior at adult stage, at which we had observed behavioral deficits in the $otpa^{-/-}$ mutant. Ablation efficiency as well as lack of collateral damage to the surrounding tissue was demonstrated by TUNEL staining to monitor apoptosis (*Figure 8A–C*). In addition, to examine the extent of recovery of cells and/or EGFP in the *Tg(oxt:egfp)* reporter, we performed a three-day follow-up imaging of ablated fish (*Figure 8D,E*). We observed a 64% mean reduction in the number of cells following ablation (*Figure 8F*).

Next, we subjected ablated and control animals to social preference and open field behavioral paradigms. The analysis showed that ablation of PT OXT neurons affects social preference (*Figure 8G*). Whereas non-ablated fish gradually habituated to the arena, as evident by the increasing time they spent in the social zone, the OXT-ablated fish displayed decreased shoal preference (*Figure 8G*). No differences were found between the groups in other tested parameters, such as the time they spent in the non-social zone or the total distance they covered (*Figure 8H,I*). The ablated

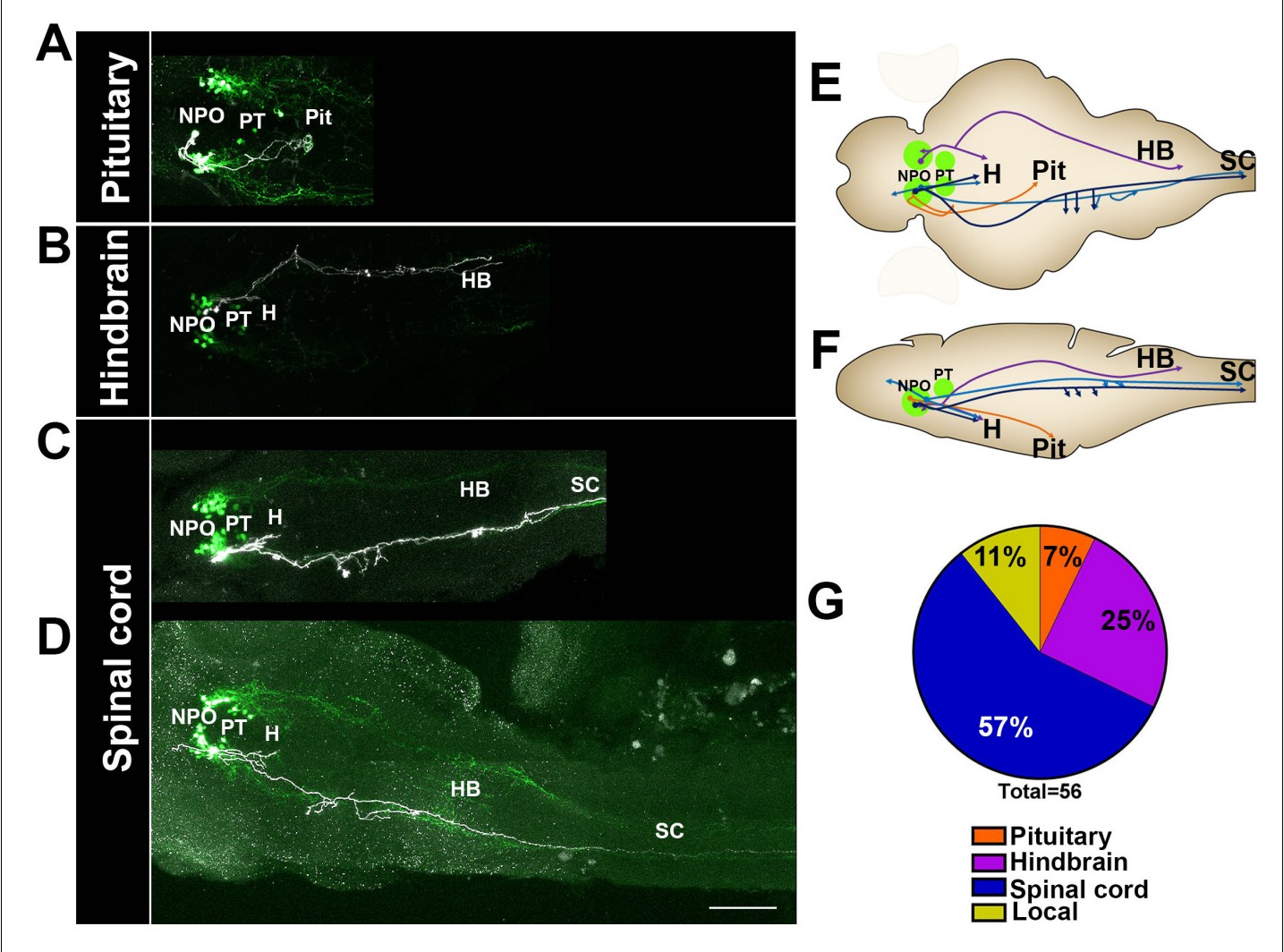

**Figure 6.** Single-cell projection mapping of anterior (NPO) OXT-ergic trajectories. (A–D) Confocal images showing representative single-cell genetic labeling of the NPO OXT neurons (grey scale) in the context of the global OXT-ergic population in a *oxt:egfp* reporter (green). In short, the *Tg(oxt: egfp)* reporter line was microinjected with OXT-specific Gal4 driver, (*oxt:Gal4*) construct together with constructs harbouring membrane localized caax-tRFP under the control of a multimerized Gal4 upstream activation sequence (*10xUAS*). Neuronal projections of RFP-labelled cells were traced and reconstituted from the 3D confocal Z-Stack. Examples of single-cell labeling of OXT projections to the posterior pituitary/neurohypophysis (A; Pit), hindbrain (B; HB) and spinal cord (C, D; SC) as well as local projections to the hypothalamus (labelled '*H*' in panels C and D) are shown. Scale bar, 100 μm. (E,F) Schemes illustrating dorsal (E) and lateral (F) views of the traced projecting neurons. (G) Pie chart showing the percentage of each type of projecting OXT neurons, which reside in the NPO.

The following figure supplement is available for figure 6:

**Figure supplement 1.** Single-cell labelling and projection tracing of OXT neurons.

fish displayed no significant alteration in novelty stress response measured by the open field assay, suggesting that this OXT cluster is mainly associated with modulation of social preference (*Figure 8—figure supplement 1*).

Taken together, these results show that developmental lesions in the PT cluster of OXT neurons have a long-term effect on adult social behavior. The early perturbation of the OXT neuronal circuit may account for the social deficit seen in the *otpa* mutants. Moreover, the observation that ablation of PT OXT neurons did not affect anxiety-like behavior uncouples the contribution of this cluster to social behavior from the general *otpa*$^{-/-}$ neuroanatomical and behavioral deficits.

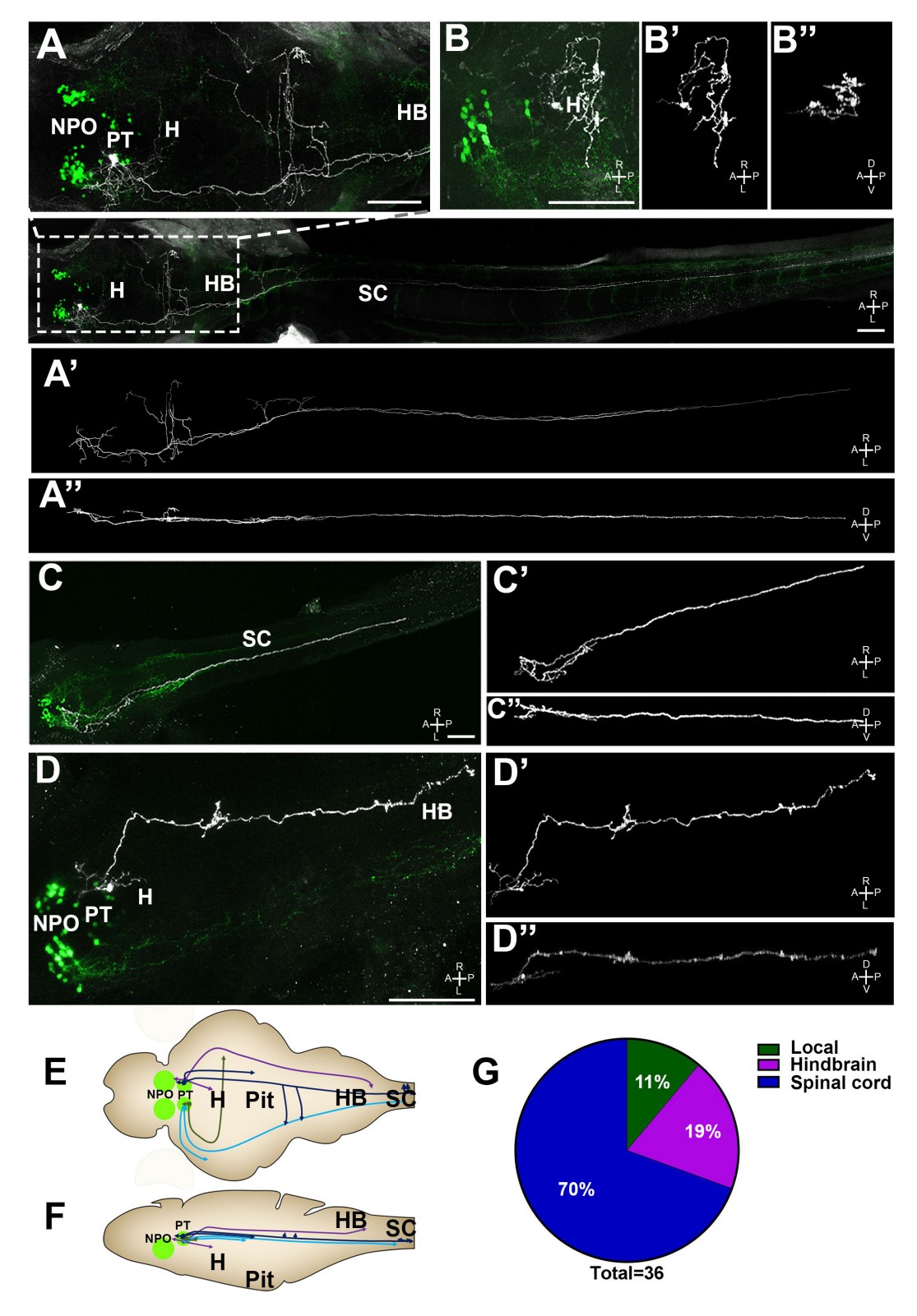

**Figure 7.** Single-cell projection mapping of posterior tuberculum (PT) OXT-ergic trajectories. (A–D) Confocal images showing representative single-cell genetic labeling of the PT OXT neurons (grey scale) in the context of the global OXT-ergic population in a *oxt:egfp* reporter (green). The 3D-traced projecting neurons are presented in **A′–D′** (dorsal) and **A″–D″** (lateral). Examples of OXT projections to the spinal cord (SC; **A** and **C**), local (**B**) and hindbrain (HB; **D**) projecting neuron are presented. The fish orientation is indicated at the bottom right corner of each image: A, anterior; D, dorsal; L,

*Figure 7 continued on next page*

*Figure 7 continued*

left, P, posterior, R, right; V, ventral. Scale bars, 100 μm. (**E,F**) Schemes illustrating dorsal (**E**) and lateral (**F**) views of the traced projecting neurons. (**G**) Pie chart showing the percentage of each type of projecting OXT neurons, which reside in the PT.

The following figure supplement is available for figure 7:

**Figure supplement 1.** Various types of projecting OXT neurons.

## Discussion

Defects in hypothalamic development may have severe consequences on the animal's ability to regulate homeostasis (*Biran et al., 2015*). Yet, the exact mechanisms by which developmental factors affect hypothalamic functions are largely unknown. In the present study we link between a critical neuroendocrine transcription factor controlling a discrete neuronal circuit and long-term developmental influence on adult behavior. The evolutionarily conserved transcription factor Otp is essential for the development of hypothalamic neurons and its embryonic knockout in mammals leads to early postpartum lethality (*Acampora et al., 1999*; *Wang and Lufkin, 2000*). Taking advantage of the viability and fertility of zebrafish with a single mutation in either of the two paralogous genes, *otpa* and *otpb* (*Fernandes et al., 2013*; *Amir-Zilberstein et al., 2012*), we examined the consequences of developmental mutations in Otp proteins on adult physiological functions. Our analysis reveals defects in anxiety- and social-related behavioral responses in adult mutants, which were associated with Otp-dependent developmental neuropeptide switching in a subset of spinal cord-projecting parvocellular OXT neurons. Specific ablation of this OXT cluster at embryonic stage resulted in reduced shoaling behavior, phenocopying the social deficits of the *otpa* mutant fish. Our study identifies a new role for Otp in regulating developmental neuropeptides switching in a discrete OXT neuronal circuit, whose developmental perturbation influences adult social behavior.

### Effects of developmental mutation in *otp* on anxiety and social behaviors

Otp controls the migration and differentiation of diencephalic neurons that populate the paraventricular nucleus (PVN) as well as the medial amygdala (MeA) (*García-Moreno et al., 2010*; *Wang and Lufkin, 2000*; *Acampora et al., 1999*). These forebrain regions are associated with the modulation of stress response and social affiliation (*Johnson and Young, 2015*; *Knobloch et al., 2012*; *Shemesh et al., 2016*). Consistently, we have found that zebrafish *otpa* mutants display deficits in stress-related response to a novel environment as well as in social shoaling preference.

We have previously shown that *otpa* mutant fish display impaired activation of the hypothalamic-pituitary-adrenal axis as well as abnormal swimming patterns in the so called 'novel tank diving' stress paradigm (*Amir-Zilberstein et al., 2012*). However, that assay measures the vertical preference toward the bottom of the test tank within the first 2 min of exposure to the arena, a tendency that is reduced to approximately chance levels by the end of a 6 min test. In the present study, we used the open field test and observed longer-lasting responses. Thus, during the entire test period, the *otpa* (but not *otpb*) mutant fish did not habituate to the novel arena and did not adopt the swimming patterns seen in wild types. The mutants display swimming characteristics that may indicate an anxiety-like response, including high rate of freezing. (*Figure 1* and *Figure 1—figure supplement 1*).

With regard to visually mediated social preference, *otpa* mutants spend less time in proximity to a shoal. The association between the increased OXT in the PT of these mutants with reduced shoaling is counter intuitive and may include modulation of other neurotransmitters. Having said that, we wish to emphasize that the OXT (and associated behavioral) phenotype presented in our study is clearly due to early developmental abnormality and not due to a classical neuro-hormonal modulatory effect. To the best of our knowledge, the long-term physiological consequence of early developmental OXT imbalance on adult behavior has not been described before. The underlying mechanism is yet to be determined.

Notably, in social animals there is a strong association between the response to a stressful experience and the social environment (*Barrett et al., 2015*; *Burkett et al., 2016*; *Smith and Wang,*

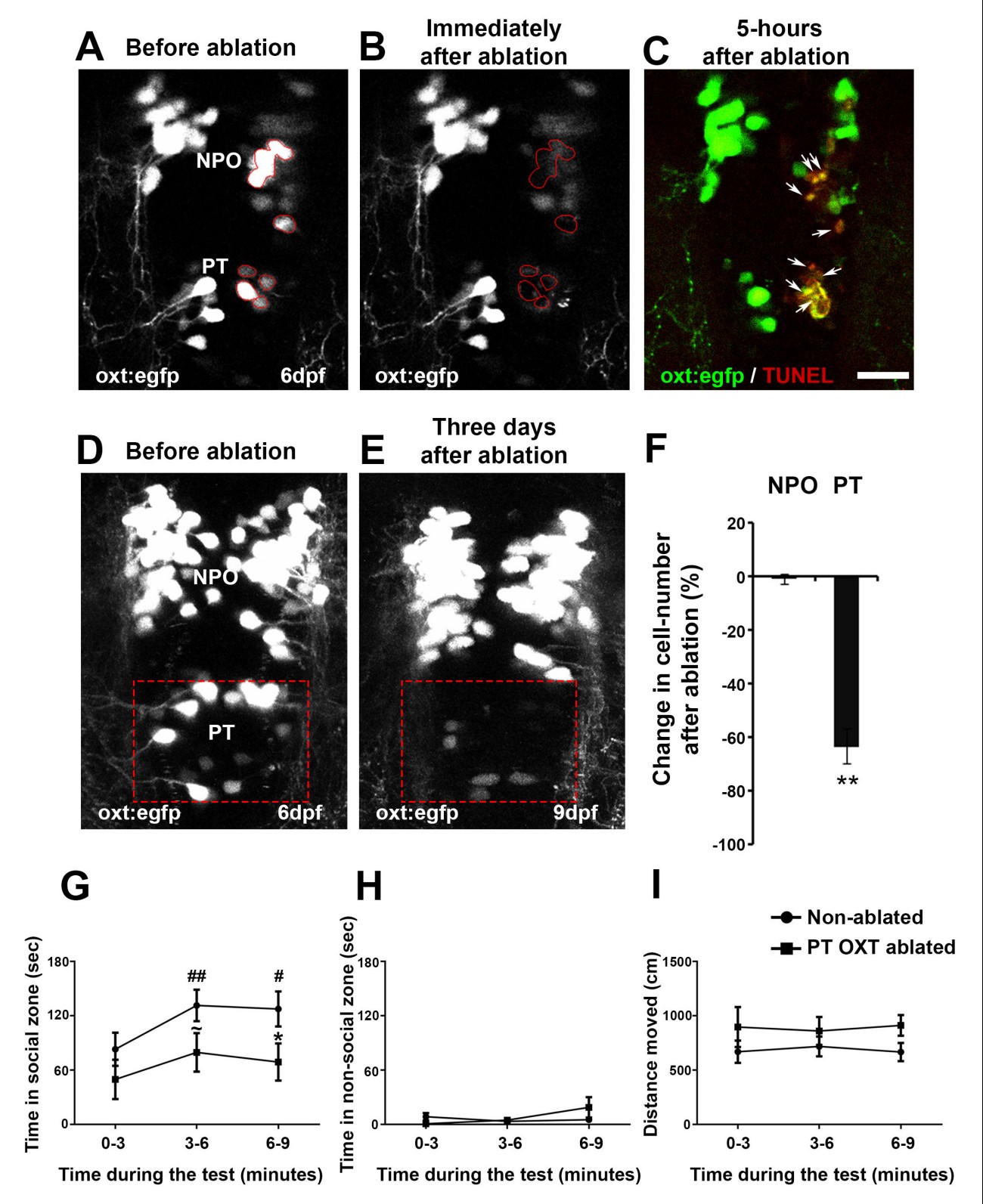

**Figure 8.** Ablation of posterior tuberculum (PT) OXT neurons affects social preference. (**A–C**) Unilateral ablation of oxt:egfp cells using two-photon microscope. Region of interest for the targeted ablation of individual cells is outlined in red (**A,B**). Specimens were fixed 5 hr after ablation and were subjected to TUNEL staining (red) to detect apoptosis and verify lack of collateral damage to the surrounding cells/tissue (**C**). The ablated OXT neurons are marked by arrows (**C**). (**D–F**) Representative images taken before (**D**) and three days after (**E**) ablation of PT OXT neurons (marked by a red

*Figure 8 continued on next page*

*Figure 8 continued*

rectangle). (F) Percentage of the change in OXT cell number after the ablation procedure in the PT region and in the non-ablated NPO area (*n* = 5). Paired sample t-test indicated that the reduction in number of OXT cells in the PT was significant [$t_{(4)}$=3.738; **p=0.010] but not in the NPO [$t_{(4)}$=0.559; p=0.303]. (G–I) Graphs showing the analysis of visually mediated social preference (VMSP) test (see schemata in *Figure 1E*) comparing the place preference of PT OXT-ablated (*n* = 12) to non-ablated (*n* = 14) zebrafish. The time spent swimming in the social zone (G), non-social zone (H) and general locomotion (I; '*distance moved*') were analyzed. (G) When comparing the '*time in social zone*' between PT OXT ablated fish and non-ablated control, there is a trend for a difference in the second time bin (~p = 0.069) and a significant difference in the third time bin (*p=0.050). In addition, while the non-ablated group spends increasingly more time in the '*social zone*' when compared to the first time bin (there is a significant main effect for 'time' in the non-ablated fish group (p=0.012); ##p≤0.01 for the 1st bin and #p≤0.05 for the second bin), no significant effect for 'time' was found for the PT OXT-ablated group (p=0.166). (H) Time spent in the '*non-social zone*' did not differ between the groups [$F_{(1,24)}$=0.212; p=0.649] and did not change throughout the test duration [$F_{(2,48)}$=2.300; p=0.135]. (I) The total distance moved in the arena did not differ between the groups [$F_{(1,24)}$=1.771; p=0.196] and did not change throughout the test duration [$F_{(2,48)}$=0.010; p=0.969].

The following source data and figure supplements are available for figure 8:

**Source data 1.** Ablation efficiency of OXT neurons and associated behavioral effects.
**Figure supplement 1.** PT OXT ablated fish habituate to an open field arena.
**Figure supplement 1—source data 1.** Swimming parameters for the open field test following OXT ablation.

*2014*; *Pagnussat et al., 2013*; *Sandi and Haller, 2015*). It has recently been shown that MeA neurons, which are associated with central control of stress, modulate the preference for novel conspecifics (*Shemesh et al., 2016*). Whether the social deficits displayed by the *otpa* mutant are associated with abnormal stress response is yet to be determined.

## Regulation of OXT by Otp paralogs

OXT is an evolutionarily conserved neuropeptide, which is involved in the modulation of social and stress behaviors in numerous species from nematodes to humans (*Wircer et al., 2016*). Our analysis of OXT expression in *otpa* and *otpb* mutant embryos revealed a previously unidentified cluster of OXT neurons in the zebrafish diencephalon, i.e. in the PT. These neurons express low levels of OXT, which is upregulated in the absence of *otpa*.

The generation of gene paralogs in zebrafish originates in genomic duplication events in teleost species (*Glasauer and Neuhauss, 2014*). The consequence of such duplications in the case of Otp is that the function of a single mammalian gene is either redundant or divided between the two paralogs, a process known as subfunctionalization. In this study, analysis of OXT neurons revealed fascinating complex genetic interactions between the two zebrafish Otp paralogs. Thus, Otpa and Otpb positively and redundantly regulate OXT expression in the NPO, as was reported by Fernandes et al. (*Fernandes et al., 2013*). In the PT, however, the two paralogs exert opposing effects on OXT cells. Otpb drives OXT expression in this region, whereas Otpa negatively regulates the expression of this neuropeptide (*Figures 2* and *4*).

We have previously demonstrated that Otp forms a complex with the *crh* promoter of zebrafish and mouse (*Amir-Zilberstein et al., 2012*). However, we were unable to demonstrate that Otp proteins directly bind to *oxt* genomic regulatory elements, suggesting that the regulation of OXT expression by Otp paralogs is indirect.

## Neuropeptide switching in Otp mutants

The ability of neurons to change their neurotransmitter repertoire has been known for some time (*Spitzer, 2015*). It has been shown that such plastic changes in peptide co-expression may allow dynamic adaptation to changes in the environmental conditions and, consequently, affect organism behavior (*Dulcis et al., 2013*; *Swanson, 1991*). We show here that a subset of OXT neurons co-express *crh*, in agreement with previous studies (*Herget and Ryu, 2015*; *Sawchenko et al., 1984*). CRH-OXT co-expressing cells exist in mammals in subpopulations of the PVN parvocellular OXT

neurons, subsets of magnocellular neurons in the PVN and SON (*Dabrowska et al., 2013*; *Sawchenko et al., 1984*), and in a small number of cells in the zebrafish NPO [(*Herget and Ryu, 2015*) and this manuscript]. This co-expression of different neuropeptides can provide functional flexibility (*Watts, 1996*).

We found that the two Otp paralogs control differential and spatially-dependent neuropeptide balance in OXT neurons (*Figure 5I*). Our finding that the developmental OXT/CRH neuropeptide switching in *otpa*$^{-/-}$ mutant is associated with later stress and social deficits is of particular interest. Notably, the involvement of OXT in stress coping and in the regulation of the hypothalamo-pituitary adrenal (HPA) stress axis, the main CRH output, has been extensively studied (*Neumann, 2002*). Furthermore, a reciprocal role of the central CRH receptor-mediated stress system in coping with social challenges has been recently shown (*Shemesh et al., 2016*). Whether the Otp-dependent regulation of OXT/CRH switching reported herein is the direct cause for the *otpa* mutant's behavioral deficits needs to be further investigated.

## Mapping of Otp-affected OXT projections

In all vertebrate species, OXT neurons affect physiological and behavioral processes, including social interactions (*Wircer et al., 2016*). The prevailing notion is that modulation of behavior by OXT is achieved by means of elaborate axonal and dendritic projections, which connect with multiple brain regions whereas neurohemal activities (e.g. reproductive physiology) are mediated via a neuroendocrine interface with fenestrated blood capillaries in the posterior pituitary (*Dölen et al., 2013*; *Knobloch et al., 2012*; *Gutnick et al., 2011*). The innervation patterns found in fish are similar to those found in mammals and OXT fibers are spread throughout the brain and spinal cord (*Saito et al., 2004*; *Goodson et al., 2003*; *Van den Dungen et al., 1982*). Yet, the exact analogy between the mammalian and zebrafish OXT clusters is not clear.

To better understand the neuroanatomy of zebrafish OXT neurons, affected by Otp, we mapped the axonal projections at the resolution of a single-cell. We found that zebrafish OXT neuron populations display highly varied innervation patterns. In particular, neurons of the NPO are structurally distinct from PT neurons. Thus, NPO OXT neurons project into the diencephalon, telencephalon, hindbrain, spinal cord and the posterior pituitary (a.k.a. the neurohypophysis) and many of them have lateral branches that project into the midbrain and hindbrain.

In contrast, none of the PT OXT neurons have hypothalamo-neurohypophyseal projections. By measuring the size of OXT neurons we found that adult zebrafish PT OXT neurons are comparable to the previously described parvocellular OXT neurons. We recently proposed, based on the expression patterns of hypothalamic markers such as Otp and Sim1, that the PT should be included as part of teleostian hypothalamus (*Biran et al., 2015*). The mammalian parvocellular OXT neurons of the PVN innervate the spinal cord and hindbrain (*Sawchenko and Swanson, 1982*; *Eliava et al., 2016*; *Swanson et al., 1980*). Thus it is possible that our newly discovered PT OXT neurons represent a part of the evolutionary origin of mammalian PVN OXT neurons.

The majority of the zebrafish PT OXT neurons contain elaborate dendritic arbors and sends long descending projections to the hindbrain and spinal cord with collateral branching to the hypothalamus, the caudal tuberculum and the tegmentum. (*Figure 7A* and *Figure 7—figure supplement 1*). Such neuroanatomy may indicate lateral modulation, which can be related to the control of motor activity (*Drapeau et al., 2002*). Interestingly, innervation of tegmental regions by OXT neurons was found to be involved in the regulation of social interactions in mammals, probably by interacting with the dopaminergic system (*Groppe et al., 2013*; *Shahrokh et al., 2010*; *Insel and Young, 2001*; *Pedersen et al., 1994*).

Spinal cord-projecting OXT neurons have been described in other species. Neurons of this type were shown to activate autonomic neurons that control penile erection in rats (*Véronneau-Longueville et al. (1999)*. This function seems to be conserved in invertebrates, in which cerebral ganglia neurons expressing the cone snail homolog for OXT/AVP, conopressin, project into the penis nerve and regulate reproductive behavior (*Van Kesteren et al., 1995*; *Wircer et al., 2016*).

A recent study performed in rats identified a small group of parvocellular OXT neurons in the PVN with collaterals in the supraoptic nucleus (SON) and spinal cord. These neurons are involved in pain relief by directly inhibiting sensory spinal cord neurons and affecting OXT release into the periphery by SON magnocellular neurons (*Eliava et al., 2016*). As most of the zebrafish PT OXT neurons project to the hindbrain and spinal cord and are characterized by an elaborate dendritic tree

and contralateral branches, we suspect that these neurons may be involved in the integration of sensory information and execution of motor output, possibly related to social, sexual or feeding behaviors and perhaps modulate response to aversive or painful stimuli (*Eliava et al., 2016*; *Rojas-Piloni et al., 2010*; *Véronneau-Longueville et al., 1999*; *Sabatier et al., 2013*).

### Uncoupling the contribution of PT OXT cluster from the general Otp behavioral effects

Otp might plays a key role in the ontogeny of social behavior and stress response by the coordination of neuropeptide repertoire in OXT neurons. The PT OXT cluster represents a relatively small and spatially discernable OXT cluster, which displays a robust *otpa*-dependent phenotype and projects predominantly to the hindbrain and spinal cord. Interestingly, we found that early developmental perturbation of the PT OXT neuronal cluster produced a specific long-term effect on their functionality leading to specific defect in the shoaling behavior, but not in the anxiety-like response (*Figure 8* and *Figure 8—figure supplement 1*). This long-term effect on social behavior might be due to changes in connectivity and/or gene expression repertoire.

We hypothesize that the behavioral deficits observed in $otpa^{-/-}$ are mainly due to the mutant's hypothalamic developmental impairments. The social behavior effect induced by the ablation procedure was less profound than the one observed in the *otpa* mutants. This may imply that the ablation affects behavior by a different mechanism. In this respect, the decreased social preference in *otpa* mutants might be influenced by other hypothalamic neurons (*Fernandes et al., 2013*; *Amir-Zilberstein et al., 2012*; *Fernandes et al., 2012*; *Blechman et al., 2007*; *Ryu et al., 2007*). Yet, we were able to demonstrate that a subtle developmental change in a small group of OXT neurons leads to a long-lasting effect on adult behavior.

## Materials and methods

### Zebrafish lines and maintenance

Zebrafish were raised and bred according to standard protocols. $Otpa^{m866}$ mutant was kindly provided by Prof. Wolfgang Driever (University of Freiburg). $Otpb^{sa115}$ mutant was generated and provided by the Sanger institute (Zebrafish Mutation Project, RRID:SCR_006161).

### Single-cell labeling and neurite tracing

*Tg(oxt:egfp)* embryos at one-cell stage were micro-injected with plasmids encoding *10xUAS:tRFP* and *oxt:Gal4* at concentration of 14 ng/μl each and with transposase mRNA at concentration of 20 ng/μl (~500 pl/embryo). At five dpf, embryos were collected and fixed in 4% PFA. Following immunostaining, embryos were mounted in 75% glycerol and scanned for tRFP-labeled OXT neurons by confocal microscopy. Using this method, we were able to attain OXT-ergic cell labeling in ~3–5% of the surviving embryos.

### Genotyping

DNA for genotyping was obtained from clipped fins of adult fish, whole embryos or from fixed samples after staining. The genomic region of interest was amplified by PCR and sequenced. The following primers were used: $otpa^{m866}$ (NM_001128703.1): sense 5'GGTCACAGGGAGGCATTAAA3', antisense 5'CGTTAAGCTGAGCCGGAGTA3'; $otpb^{sa115}$ (BC076366.1): sense 5'GTCCACAGGGATGAAGGATG3', antisense 5'GTCCTGTGGCGTTTCTGTTT3'.

### In situ hybridization and immunostaining

RNA in situ hybridization was performed as described in *Machluf and Levkowitz (2011)*. For probe preparation, pGem plasmids encoding for *oxt* mRNA (RefSeq NM_178291.2) or *crh* mRNA (provided by Giselbert Hauptmann). *otpb* (NM_131100), *otpa* (NM_001128703.1) and *oxtr* (NM_001199370.1) probes were synthesized from a PCR-based template using the following primers: *otpb*: CACTACAAACCTCAAGTATTC; CCCACTTAACAATCATTG, *otpa*: 5'CAGTGTCCATGAGCTTCAC3'; CGAGTGCACCTTGTTTCT and *oxtr*: TGATTGCTGGGGGAGATTTTGTTCA; TTATGTGATGGAGGTTTGGGTGA. Single molecule *oxt* mRNA detection was done with Stellaris probes as describes in (*Orjalo et al., 2011*).

Immuno-fluorescent staining was done as described in the Zebrafish Brain Atlas (RRID:SCR_000606) (http://zebrafishbrain.org/protocols.php) using the following primary antibodies: Chicken anti-EGFP (A10262; Life technologies/Thermo Fisher, Waltham, MA USA), rabbit anti-tRFP (AB234; Evrogen, Moscow, Russia) and mouse anti-TH (MAB318 clone LNC1; Milipore-Chemicon, Billerica, MA). Secondary antibodies were purchased from Jackson ImmunoResearch Laboratories (West Grove, PA).

## Image acquisition and analysis

Colorimetric images were obtained using X 20 objective on a Zeiss Axioplan microscope (Zeiss, Jena, Germany). Images of fluorescently labeled samples were obtained by using Zeiss LSM 710 confocal microscope with oil immersion X 40 lenses. Images were analyzed using the open source FIJI image-processing package. Neurites were traced using the Simple Neurite Tracer plugin. Cell numbers were counted using the Cell Counter plugin.

## Laser ablation and two-photon scanning microscopy

Photo-ablation of EGFP-positive neurons was performed in 6-day-old larvae in 30% Danieau's solution without methylene blue. Embryos were anesthetized with Tricaine (MS-222; Sigma-Aldrich, St. Louis, MO) and were mounted in Difco Agar Noble (BD, Sparks, MD). We used LSM7 multi-photon (MP) Laser scanning microscope (Zeiss, Jena, Germany) with modified Achroplan X 40 0.8 W, NA 1.0, for both imaging and ablation. For the ablation laser was set on 920 nm, 100% power for 20 iterations directed at EGFP-labeled cells to generate photo-induced singlet oxygen-mediated apoptosis without damaging the surrounding cells/tissue. In order to detect apoptotic cells, embryos were fixed overnight in 2% PFA at 4°C and stored at 1% PFA until staining.

## Terminal deoxynucleotidyl transferase dUTP nick end labeling (TUNEL) staining

Detection of apoptotic cells was done by using the ApopTag Red In Situ Apoptosis Detection Kit (Millipore, Temecula, CA).

## Behavioral assays

### Video acquisition

Behavioral assays were performed using a custom-made apparatus. The fish were placed in designated tanks on top of a light table and illuminated from below, using infrared wavelengths (intensity peak, 875 nm). The camera was positioned above the table and the fish were filmed through an optical cast infrared longpass filter (Edmund Optics, Barrington, NJ). Videos were acquired with a 2M360-CL camera (IO Industries, London, Ontario), with an image acquisition Sapera LT-development package (Teledyne Dalsa, Waterloo, Ontario) and recorded with Stream5 software (IO Industries, London, Ontario). Behavior recording was done with EthoVision video tracking system (Noldus Information Technologies, Wageningen, The Netherlands). Relevant data was exported into Excel for further analysis.

### Open field

Fish were placed in a circular arena of 23 cm in diameter filled to a height of 5 cm with regular system water. Swimming was recorded for 10 min.

### VMSP

The visually mediated social preference (VMSP) test was developed based on *Engeszer et al. (2007)*. In this test, a single fish was placed for 5 min in a transparent 'start box' in a rectangular arena (20.3 × 19.4 × 5 cm) from which two compartments (each sized 14.4 × 8.5 × 5 cm), separated from each other by an opaque partition were visible: one containing a four-fish shoal and the other empty. The stimuli were randomized in these two compartments, to avoid a side bias. After an acclimatization period, the focal fish was released from the start box and allowed to explore the arena, and its behavior was video-recorded for 9 min for subsequent analysis. All compartments were separated by sealed transparent partitions, avoiding the access to olfactory stimuli. The time spent by

the focal fish close (less that one body length) to each compartment (termed social zone or non-social zone) was quantified and taken as a measure of social preference.

## Statistical analyses

Data is presented as mean ± standard error of the mean (SEM) and analyzed using SPSS 20.0. All data sets were tested for departures from normality with Shapiro-Wilks test. Students t-test or Mann-Whitney was used for all comparisons between two groups. ANOVA or Kruskal-Wallis H test (when samples departed from normal distribution) were used for comparing multiple groups. Two factor univariate ANOVA was used when necessary. All data sets were corrected for multiple comparisons. Dunn's pairwise comparisons, student t-test and Bonferroni comparisons were used as post-hocs. * indicates p<0.05, ** indicates <0.01 and *** indicates p<0.001.

## Statistical analysis used in *Figure 1*

In order to assess the effects of mutations in the *otp* genes on anxiety, the fish swimming was measured in an open field. The fish anxiety-like behavior, as indicated by changes in 'speed' and 'distance from wall' was analyzed by two-way ANOVA for 'genotype' [between subject factor (wild type/ *otpa*$^{-/-}$/ *otpb*$^{-/-}$)], 'time' during the test [within subject factor with repeated measures (minutes 1–10)] and their interaction ('genotype' × 'time'). These analyses indicated the following:

'Speed'- significant main effects for 'time' [$F_{(9,405)}$=17.571; p=0.000] and for 'genotype' [$F_{(2,45)}$=10.380; p=0.000]; the interaction 'genotype' × 'time' was also significant [$F_{(9,405)}$=5.737; p=0.000]. Scheffe post-hoc analyses indicated that *otpa*$^{-/-}$ mutants differ significantly from both wild types (p=0.000) and *otpb*$^{-/-}$ mutants (p=0.020); wild types and *otpb*$^{-/-}$ mutants did not differ (p=0.447). Follow-up analyses evaluated the main effect of 'time' with-in each of the 'genotype' groups; a significant main effect for 'time' was found in both the wild types and *otpb*$^{-/-}$ groups [wild type: $F_{(9,99)}$ = 8.712; p=0.000; *otpb*$^{-/-}$: $F_{(9,108)}$ = 11.816; p=0.000], but not in the *otpa*$^{-/-}$ group [$F_{(9,198)}$=2.964; p=0.771].

'Distance from wall' - significant main effects for 'time' [$F_{(9,405)}$=29.914; p=0.000] and for 'genotype' [$F_{(2,45)}$=26.545; p=0.000]; the interaction 'genotype' × 'time' was also significant [$F_{(9,405)}$=3.004; p=0.002]. Scheffe post-hoc analyses indicated that *otpa*$^{-/-}$ mutants differ significantly from both wild types (p=0.000) and *otpb*$^{-/-}$ mutants (p=0.000); wild types and *otpb*$^{-/-}$ mutants did not differ (p=0.988). Follow-up analyses evaluated the main effect of 'time' with-in each of the 'genotype' groups; a significant main effect for 'time' was found in all the groups [wild type: $F_{(9,99)}$ = 25.079; p=0.000; *otpb*$^{-/-}$: $F_{(9,108)}$ = 22.057; p=0.000; *otpa*$^{-/-}$: $F_{(9,198)}$ = 3.326; p=0.016], however contrast comparisons (comparing each minute to First minute; corrected for repeated measures) indicated that in both wild types and *otpb*$^{-/-}$ the fish swam closer to the wall as of the second minute, while *otpa*$^{-/-}$ mutants did so only as off the eighth minute.

The fishes 'social preference' (time spent in the 'social' section of the arena) was analyzed by two-way ANOVA for 'genotype' [between subject factor (wild type/ *otpa*$^{-/-}$/ *otpb*$^{-/-}$)], 'time' during the test [within subject factor with repeated measures (3 min bins)] and their interaction 'genotype' × 'time'. These analyses indicated a significant main effect for 'genotype' [$F_{(2,27)}$=4.237; p=0.025] and a 'near significant' main effect for 'time' [$F_{(2, 54)}$=2.790; p=0.070]; the interaction 'genotype' × 'time' was also significant [$F_{(2,54)}$=4.284; p=0.004]. Further ANOVA comparisons per time bin indicated no difference between the groups in the 1st time bin [$F_{(2)}$=2.886; p=0.073], but a significant differences between the groups in both the second [$F_{(2)}$=5.373; p=0.011] and third [$F_{(2)}$=4.290; p=0.024] time bins; Scheffe post-hoc analyses indicated that *otpa*$^{-/-}$ fishes spent significantly less time in the social zone from both wild types (second bin: p=0.049; third bin: p=0.047) and *otpb*$^{-/-}$ (second bin: p=0.019; third bin: p=0.068); wild type and *otpb*$^{-/-}$ did not differ (second bin: p=0.914; third bin: p=0.984).

Similar analyses of time spent in the 'non-social' zone indicated no significant main effects; 'genotype' [$F_{(2,27)}$=2.600; p=0.093]; 'time' [$F_{(2, 54)}$=1.385; p=0.258]; the interaction 'genotype' × 'time' was not significant [$F_{(2,54)}$=0.254; p=0.851].

No differences were observed between the genotypes in their home tank locomotion in either percent of time spent moving [$X^2_{(2)}$=3.605; p=0.165] or their swimming speed [$F_{(2)}$=1.293; p=0.288].

## Statistical analysis used in *Figure 4*

ANOVA indicted a significant main effects for 'genotype' on the number of *oxt* expressing neurons in the NPO [$F_{(8)}$=36.361; p=0.000]; Dunnett post-hoc analyses indicated that *otpa*$^{+/+}$;*otpb*$^{+/+}$ differed significantly only from *otpa*$^{-/-}$;*otpb*$^{+/-}$ (p=0.013) and *otpa*$^{-/-}$;*otpb*$^{-/-}$ (p=0.000). Kruskal-Wallis indicated a significant main effects for 'genotype' on the number of OXT expressing neurons in the PT [$X^2_{(8)}$=85.074; p=0.000]; Dunn's corrected pair-wise comparisons indicated that the *otpa*$^{+/+}$;*otpb*$^{+/+}$ differed significantly (p=0.000) only from *otpa*$^{-/-}$;*otpb*$^{+/-}$ and *otpa*$^{-/-}$;*otpb*$^{+/+}$.

## Statistical analysis used in *Figure 8*

One sample t-test indicated a significant reduction in the number of OXT expressing neurons in the PT [$t_{(4)}$=3.738; p=0.010] but not in the NPO [$t_{(4)}$=0.559; p=0.303].

The fish 'social preference' (time spent in the 'social' section of the arena) was analyzed by two-way ANOVA for 'ablation' [between subject factor ('non-ablated'/ 'PT OXT ablated')], 'time' during the test [with-in subject factor with repeated measures (three time bins)] and their interaction 'ablation' × 'time'. These analyses indicated a significant main effect for 'time' [$F_{(2,48)}$=8.138; p=0.003] and a near significant main effect for 'ablation' [$F_{(1,24)}$=3.591; p=0.070]; the interaction 'ablation × 'time' was not significant [$F_{(2,48)}$=0.810; p=0.416]. Follow-up analyses evaluated the main effect of 'time' [within subject factor with repeated measures (three time bins)] within each of the 'ablation' groups. These analyses indicated a significant main effect for 'time' in the 'non-ablated' group [$F_{(2,26)}$=7.249; p=0.012], but not in the 'PT OXT ablated' group [$F_{(2,22)}$=2.031; p=0.166]. Contrast comparisons (corrected for repeated measures) indicated that in the 'non-ablated' group the time spent in the social section of the arena increased significantly throughout the test duration; as compared with the 1st time bin the fish spent significantly more time in the social section of the arena during the second and third bins [second bin: $F_{(1,13)}$ = 9.928; p=0.008. third bin: $F_{(1,13)}$ = 6.365; p=0.025]. Further t-tests comparisons per time bin indicated no difference between the groups in the 1st time bin [$t_{(24)}$=1.179; p=0.250], a trend for a difference in the second time bin [$t_{(24)}$=1.903; p=0.069] and a significant difference in the third time bin [$t_{(24)}$=2.069; p=0.050].

## Statistical analysis used in *Figure 8—figure supplement 1*

In order to assess the effects of PT OXT ablation on anxiety, the fish swimming was measured in an open field test. The fish anxiety-like behavior, as indicated by their 'speed' and 'distance from wall' while swimming, was analyzed by two-way ANOVA for 'ablation' [between subject factor ('non-ablated'/ 'PT OXT ablated')], 'time' during the test [within subject factor with repeated measures (minutes 1–10)] and their interaction ('ablation' × 'time'). These analyses indicated the following: In 'distance from wall' there was a significant main effect only for 'time' [$F_{(9,243)}$=13.905; p=0.000]; both 'ablation' [$F_{(1,27)}$=0.449; p=0.508] and the interaction 'ablation' × 'time' [$F_{(9,234)}$=1.344; p=0.263] were not significant. Follow-up analyses evaluated the main effect of 'time' within each of the groups; a significant main effect for 'time' was found in both groups ['non-ablated': $F_{(9,135)}$ = 10.635; p=0.000; 'PT OXT ablated': $F_{(9,108)}$ = 4.748; p=0.006].

There were a significant main effects for 'time' [$F_{(9,243)}$=6.339; p=0.000] and for 'ablation' [$F_{(1,27)}$=4.470; p=0.044] on swimming speed; the interaction 'ablation' × 'time' was not significant [$F_{(9,243)}$=1.621; p=0.185]. Follow-up analyses evaluated the main effect of 'time' within each of the 'ablation' groups; a significant main effect for 'time' was found in the 'non-ablated' group [$F_{(9,135)}$=6.468; p=0.0011], but not in the 'PT OXT ablated' group [$F_{(9,108)}$=1.65; p=0.201]. Contrast comparisons (comparing each minute to First minute; corrected for repeated measures) indicated that in the 'non-ablated' group the fish swam faster as of the fourth minute, while 'PT OXT ablated' fishes did so as of the fifth minute.

## Acknowledgements

We thank Raya Eilam for assisting with the TUNEL staining; Wolfgang Driever for kindly providing the otpa$^{m866}$ mutant line; The Sanger Institute for providing the *otpb*$^{sa115}$ mutant line; Chi-Bin Chien for the Tol2kit plasmid vectors; Giselbert Hauptmann for the *crh* probe; Michael Gliksberg and Eva Mishor for establishing the behavioral test settings in the Levkowitz lab; Asif Wircer for the brain illustrations; Nitzan Konstantin for English editing. RN was supported by the Weizmann's Dean of

Faculty postdoctoral fellowships. The research in the Levkowitz lab is supported in part by the Adelis Metabolic Research Fund, (in the frame of the Weizmann Institute). GL is an incumbent of the Elias Sourasky Professorial Chair.

## Additional information

### Funding

| Funder | Grant reference number | Author |
|---|---|---|
| Israel Science Foundation | 1511/16 | Einav Wircer<br>Janna Blechman<br>Gil Levkowitz |
| Israel Science Foundation | 957/12 | Einav Wircer<br>Janna Blechman<br>Nataliya Borodovsky<br>Gil Levkowitz |
| Israel Science Foundation | 2137/16 | Janna Blechman<br>Gil Levkowitz |

The funders had no role in study design, data collection and interpretation, or the decision to submit the work for publication.

### Author contributions

EW, Conceptualization, Data curation, Formal analysis, Writing—original draft, Writing—review and editing; JB, Conceptualization, Data curation, Validation; NB, Data curation, Validation; MT, Formal analysis, Preformed the statistical analysis and helped establish some of the behavioral settings; ARN, Writing—review and editing, designed and established the VMSP test contributed to the data interpretation; RFO, Writing—review and editing, Designed and established the VMSP test contributed to the data interpretation; GL, Conceptualization, Supervision, Funding acquisition, Writing—original draft, Project administration, Writing—review and editing

### Author ORCIDs

Rui F Oliveira, http://orcid.org/0000-0003-1528-618X
Gil Levkowitz, http://orcid.org/0000-0002-3896-1881

### Ethics

Animal experimentation: All procedures were approved by the Weizmann Institute's Institutional Animal Care and Use Committee protocol (27220516-3)

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
