## [Decision Letter]

Thank you for submitting your article "Otp affects developmental peptide switching in oxytocin neurons associated with a long-term effect on social behavior" for consideration by *eLife*. Your article has been favorably evaluated by a Senior Editor and three reviewers, one of whom is a member of our Board of Reviewing Editors. The following individual involved in review of your submission has agreed to reveal their identity: Larry Young (Reviewer #3).

The reviewers have discussed the reviews with one another and the Reviewing Editor has drafted this decision to help you prepare a revised submission.

Summary:

Wircer and colleagues describe results from a series of studies investigating the role of a hypothalamic transcription factor (Otp) in regulating oxytocin expression neurons in the zebrafish. In brief, they report that Otpa but not Otpb mutations affect neuropeptide switching, which results in alterations in anxiety-like and social behaviors in adult zebrafish.

The data suggest that Otpa, but not Otpb robustly modulates both anxiety-related behavior and shoaling behavior in zebrafish. The two paralogues appear to have redundant roles on the regulation of Oxt expression in the neurosecretory portion of OXT neurons in the NPO. However, Otpa mutants revealed a novel group of OXT neurons in the posterior tuberculum (PT) that had not been previously described. Oxt is upregulated in Otpa mutants compared to WT and Otpa mutants, revealing a unique role of the Otpa paralogue in suppressing Oxt expression in these cells. Interestingly, Otpa appears to be involved in neuropeptide switching in both NPO and PT, albeit in opposite directions. Otpa mutants have more Oxt/CRF co-expression in the NPO, but less co-expression in the PT. Notably, they report that the neurons mediating this effect are neurons that project to the brainstem and spinal cord. They used single cell analysis to reveal that the PT OXT neurons primarily project to the hindbrain and brainstem. Ablation of the PT OXT neurons developmentally resulted in reduced shoaling behavior but did not affect anxiety behaviors.

Collectively, the findings of this paper are novel and interesting and suggest an important contribution of a defined subset of OXT neurons to behaviors in the zebrafish model. Given the conservation of the oxytocinergic system across species, these insights may be applicable to higher order vertebrates, including humans.

Essential revisions:

In its current description, the interpretation of the visually mediated social preference test appears to be confounded by the findings from the open tank swim test. If *otpa* null fish display reduced habituation in the open tank test, one might expect that they would be less likely to approach the socialization zones of the preference testing tank. By the design of the preference testing, *otpa* mutants would be less likely to enter the zones of socialization. Stated another way, is the social preference tank really just another novel tank test? Habituation to and normal exploration of the preference tank should be documented before conclusions about social preference are inferred.

The molecular explanation for *otpa* being required for *oxt* transcription in the NPO yet suppressing *oxt* transcription in the PT is underdeveloped. At a minimum, colocalization of *otpa* and *otpb* within *oxt:egfp* PT neurons would add to this initial characterization. In addition, the data demonstrating the% of *crh*-positive OXT cells in the different brain regions should be represented as total number of cells (Figure 4). None of the images presented suggest that there are more than 5-8 PT OXT neurons, so using percentages of this value that express *crh* will magnify graphical differences. Moreover, is *crh* itself altered by the inactivation of *otpa* and/or *otpb*? Data from Fernandes et al. PlosOne 2013 (http://dx.doi.org/10.1371/journal.pone.0075002) suggest this to be the case. This should be clearly stated in the Discussion.

---

## [Author Response]

Essential revisions:

In its current description, the interpretation of the visually mediated social preference test appears to be confounded by the findings from the open tank swim test. If otpa null fish display reduced habituation in the open tank test, one might expect that they would be less likely to approach the socialization zones of the preference testing tank. By the design of the preference testing, otpa mutants would be less likely to enter the zones of socialization. Stated another way, is the social preference tank really just another novel tank test? Habituation to and normal exploration of the preference tank should be documented before conclusions about social preference are inferred.

We appreciate the reviewers’ concern that given the result of the open field, “*otpa* mutants would be less likely to enter the zones of socialization” due to increased anxiety. To address this concern, we analyzed the number of entries into the social zone during the social preference test and found no significant difference between wild type and *otpa* mutant animals (new Figure 1—figure supplement 1). Thus, our interpretation is that *otpa* mutants enter as much the social zone as wild type control but stay there less time. We conclude that *otpa* mutants display normal exploration of the social zone.

Regarding the question “is the social preference tank really just another novel tank test?”. As we described in the Methods section, the design of social preference test (VMSP) is different than the open field arena in two major respects: 1) Fish are subjected to overnight social isolation prior to testing. 2) fish are placed in a transparent cylinder (“start box”) located at the opposite side of the stimulus shoal for 5 minutes prior to VMSP recording. As the reviewers requested we documented the habituation to the preference arena as measured by the distance moved and speed over time (new Figure 1—figure supplement 1). This analysis showed no significant difference between *otpa* mutants and wild type fish. Thus, *otpa* mutants do not seem to have a difference in the habituation to the test tank that prevents them to express normal exploration of the social zone.

We refer to all of the above in the revised manuscript (subsection “*otpa* but not *otpb* mutants display anxiety and social behavior deficits”, third paragraph).

The molecular explanation for otpa being required for oxt transcription in the NPO yet suppressing oxt transcription in the PT is underdeveloped. At a minimum, colocalization of otpa and otpb within oxt:egfp PT neurons would add to this initial characterization.

The revised manuscript now includes the requested experiment in which we performed *in situ* hybridization showing colocalization of *otpa* and *otpb* mRNA within *oxt:egfp* PT neurons. This data is presented in a new Figure 2—figure supplement 3 and referred to in the first paragraph of the subsection “Otp paralogs exert differential effects on OXT neuronal clusters”.

In addition, the data demonstrating the % of crh-positive OXT cells in the different brain regions should be represented as total number of cells (Figure 4). None of the images presented suggest that there are more than 5-8 PT OXT neurons, so using percentages of this value that express crh will magnify graphical differences.

As requested, we now present both the percentage and total number of *crh*-positive OXT cells (Figure 5 of the revised manuscript). We also increased the sample size as we obtain more data since the submission of the paper. Both types of analyses are in accordance with our initial observations and conclusions regarding the effects of Otp paralogs on neuropeptide switching in the NPO and PT.

Moreover, is crh itself altered by the inactivation of otpa and/or otpb? Data from Fernandes et al. PlosOne 2013 (http://dx.doi.org/10.1371/journal.pone.0075002) suggest this to be the case. This should be clearly stated in the Discussion.

We thank the reviewers for the opportunity to clarify this point. The results presented in our manuscript demonstrate that the balance between *crh* and *oxt* is altered following inactivation of either *otpa* or *otpb* without affecting the total number of *crh* neurons. In fact, the results by Fernandes et al. are in full agreement with our analysis (see Figure S2 in the “Supporting Information” of Fernandes et al.):

1) No change in the total number CRH^+^ neurons in the NPO and PT following inactivation of either *otpa* or *otpb.*

2) Marked decrease in the double *otpa/otpb* mutant.

As requested by the reviewers, this is now clearly stated in the revised manuscript (subsection “Otp paralogs differentially regulate neuropeptide switching in OXT neurons”, first paragraph).